



# Simplified SAGE II ozone data usage rules

Stefanie Kremser [1], Larry W. Thomason [2], and Leroy J. Bird [1]

[1]Bodeker Scientific, Alexandra, New Zealand
[2]NASA Langley Research Center, Hampton, USA

**Correspondence:** Stefanie Kremser (stefanie@bodekerscientific.com)

**Abstract.** High quality satellite-based measurements are crucial to the assessment of global stratospheric composition change. The Stratospheric Aerosol and Gas Experiment II (SAGE II) provides to date the longest, continuous data set of vertically resolved ozone and aerosol extinction coefficients and therefore, remains a cornerstone of understanding and detecting long-term ozone variability and trends in the stratosphere. Despite its stability, SAGE II measurements must be screened for outliers

that are a result of excessive aerosol emitted into the atmosphere and that degrade inferences of change. Current methods for SAGE II ozone measurement quality assurance consist of multiple ad-hoc, sometimes conflicting rules, leading to too much valuable data that are being removed or outliers being missed. In this work, the SAGE II ozone data set version 7.00 is used to develop and present a new set of screening recommendations and to compare the output to the screening recommendations currently used. Applying current recommendations to SAGE II ozone lead to unexpected features, such as

removing ozone values around zero if the relative error is used as a screening criteria, leading to biases in monthly mean zonal mean ozone concentrations. Most of these current recommendations were developed based on 'visual inspection', leading to inconsistent rules that might not be applicable at every altitude and latitude. Here, a set of new screening recommendations is presented that take into account the knowledge about how the measurements were made. The number of screening recommendations is reduced to three, which mainly remove ozone values that are affected by high aerosol load-

ing and therefore are not reliable measurements. More data remain when applying these new recommendations compared to the rules that are currently being used, leading to more data being available for scientific studies. The SAGE II ozone data set used here is publicly available at https://doi.org/10.5281/zenodo.3710518. The complete SAGE II version 7.0 data set, which includes other variables in addition to ozone, is available at https://eosweb.larc.nasa.gov/project/sage2/sage2_v7_table, doi:10.5067/ERBS/SAGEII/SOLAR_BINARY_L2-V7.0 (SAGE II Science Team, 2012; Damadeo et al., 2013).

## 1   Introduction

Even though the stratosphere contains less than 10 % of the mass of the atmosphere, changes in its chemical composition affect surface climate. For instance, stratospheric ozone is the key factor in governing UV levels at Earth's surface, which directly impact human, animal and plant health. In addition, many stratospheric components including ozone absorb and emit radiation, which in turn change the temperature distribution within the stratosphere, and therefore change the dynamics through the

atmosphere, down to the surface. Understanding how the fingerprint of the effects of the stratosphere on the climate system changes with time is therefore imperative for diagnosing past and ongoing changes in climate. Space-based measurements of





stratospheric composition are useful for diagnosing global changes as they can provide consistent, long-term measurements of key parameters on a global scale. However, these measurements are complicated by the fact that they are indirect measurements (usually optical properties of their targets) whose quality is challenged by the physics of the measurement including

the accuracy of the measurement and the ability to separate between the effects of the target species and the affects of other gases and particulate matter. As a result, our ability to accurately detect small, but important changes in a key parameter like ozone is controlled by our ability to identify the difference between usable and deficient data. The process of making these distinctions is crucial to deriving robust long-term trends in ozone and other compounds as well as understanding the impact of large geophysical events such as the 1991 Mt Pinatubo eruption.

Measurements from the Stratospheric Aerosol and Gas Experiment II (SAGE II, McCormick (1987)) remain a cornerstone of understanding and detecting long-term ozone variability and trends in the stratosphere. This data set is recognised for its stability over its 21-year lifetime (1984-2005) and the high vertical resolution of its ozone and aerosol extinction coefficient measurements during its mission. While this instrument was remarkably successful and long lasting, it is well known that SAGE II ozone data quality declines due to the high stratospheric aerosol levels following the Mt Pinatubo eruption and other

deleterious features. As with any data set, these must be accounted for when using SAGE II data. Given that the mission began 35 years ago and the Mt Pinatubo eruption occurred almost 30 years ago, it is unsurprising that numerous filters for removing artifacts in SAGE II ozone data have been proposed (e.g. Cunnold et al., 1989), modified, and refined (Cunnold et al., 1996; Wang et al., 2002; Rind et al., 2005; Wang et al., 2006). As a result, there are a plethora of 'generally accepted' screening methods for SAGE II data that are sometimes inconsistent with one another, most often subjective, and occasionally

untraceable. Furthermore, the screening recommendations derived using one version of the SAGE II data are continued to be used with a later, and presumably better version of the SAGE II data set, but without revision of the recommendations when the data, its reported uncertainty, and its sensitivity to interfering species are likely to have changed. For instance, the release notes for version 7.0 of the SAGE II data recommend use of the Wang et al. (2002) ozone usage rules, which were developed for version 6.1 of the SAGE II data. Therefore, there remains a need for consistent and robust approaches to determining the

suitability of stratospheric chemical composition measurements to diagnose long-term change.

SAGE II ozone data are often used as reference measurements to which other measurements are adjusted (Froidevaux et al., 2015; Davis et al., 2016; Sofieva et al., 2017; Hassler et al., 2018), as they provide a long stable record of ozone measurements with a high vertical resolution. Therefore, having the best possible SAGE II ozone data set is crucial in the development of long-term homogeneous data sets that combine measurements from different sources. However, having numerous screening

recommendations available that were published in previous studies hampers the use of the SAGE II data in scientific studies and the creation of merged ozone data sets such as Sofieva et al. (2017); Froidevaux et al. (2015); Davis et al. (2016); Hassler et al. (2018). Furthermore, the impact of these recommendations on the SAGE II data set is seldom further investigated, and any introduced biases will remain undetected.

In this paper, we will use the SAGE II data set version 7.0 (Damadeo et al., 2013; SAGE II Science Team, 2012) to compare

the impact of SAGE II ozone data usage rules that have been proposed in a number of publications (Wang et al., 2002; Rind et al., 2005; Wang et al., 2006) and are stated on the SAGE II release notes page. Here, we focus on the proposed rules that





were applied to the SAGE II ozone data set before being used to generate a homogenized satellite record of vertically resolved ozone data sets (Davis et al., 2016; Hassler et al., 2008, 2018). A long-term, latitudinally and vertically resolved ozone database is required as input to climate models that do not have the ability to include a fully coupled stratospheric chemistry scheme

(Hassler et al., 2018). Both Davis et al. (2016) and Hassler et al. (2018) applied around seven rules that are mainly based on previous studies by Wang et al. (2002) with the modifications outlined in the SAGE II version 7.0 release notes.

The aim of this study is not to review the 'generally accepted' set of rules for SAGE II ozone data in detail, but to develop a set of simple yet robust SAGE II ozone data usage rules. A characteristic of any limb-viewing UV/visible instrument such as SAGE II is that the quality of any ozone observation is sensitive to material that lies at and above the altitude of the observation.

Therefore, we will modify aerosol-related rules to reflect the geometry of SAGE II ozone measurements. The rules will make use of parameters that are generally available or derivable from information contained in the SAGE II version 7.0 data files. The expectation is that these new rules to SAGE II ozone data results in a more robust SAGE II ozone data set that can be used in trend analysis studies and homogenization efforts with other space-based measurements.

## 2   SAGE II - the basics

The SAGE II instrument was on board the Earth Radiation Budget Satellite (ERBS), launched by the space shuttle Challenger in October 1984 and was operational until mid-2005. Like its companion instruments the Stratospheric Aerosol Measurement (SAM II, 1978-1993), SAGE (1979-1981), SAGE III/Meteor 3M (2002-2005), and SAGE III/ISS (2017-present), SAGE II observed the Sun through the limb of the atmosphere for each spacecraft transit of the solar terminator, a technique called solar occultation (Fig. 1), to measure a line-of-sight (LOS) transmission profile from 0.5 to 100 km at multiple wavelengths.

Each profile took between 1.5 and 4 minutes to collect with up to 32 profiles per day at its peak, although the number of profiles decreased to 16 after mid-2000. Combined with the ERBS orbit this provided measurements at two latitudes per day that shift over time, providing coverage from 80° S to 80° N every 1–2 months. In SAGE terminology, each spacecraft sunrise and sunset encounter producing these LOS transmission profiles is referred to as an 'event'; there are usually fifteen sunset and fifteen sunrise events each day. The altitude at which transmission was reported is for the lowest most point along the

LOS path and commonly referred to as the tangent altitude ($Z_t$ in Fig. 1) since it corresponds to the point where the path is traveling parallel to the Earth's surface. The lower most altitude where transmission was reported is generally higher than 0.5 km since the presence of dense cloud or aerosol and occasionally the solid Earth itself make the line of sight opaque. The geometry of the solar occultation measurement technique is shown in Fig. 1. Under most circumstances, this geometry is favorable for stratospheric applications including ozone and aerosol extinction coefficient since the long path lengths near the

tangent altitude allow for a large signal-to-noise ratio for what are generally optically thin layers of ozone and aerosol. SAGE II has a large dynamic range and can theoretically measure LOS optical depths between about 0.001 and 8. The long paths have a broad horizontal resolution with an effective horizontal resolution between hundreds and tens of thousands of square kilometers depending on the details of an individual event (Thomason and Taha, 2003). SAGE II was a seven-channel Sun photometer with central wavelengths at 386, 448, 452, 525, 600, 940 and 1020 nm. The latest algorithm used to derive version

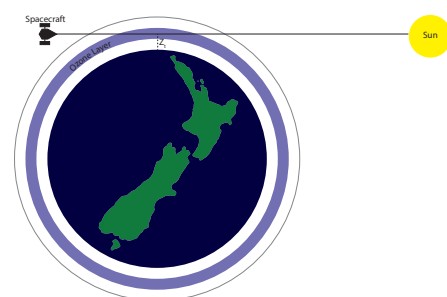

**Figure 1.** A schematic diagram showing the solar occulation viewing geometry which is exploited by SAGE II to make measurements of atmospheric constituents. $Z_t$ denotes the tangent altitude. For more details see text.

7.0 of SAGE II data is described in detail in Damadeo et al. (2013). The most challenging part of the data processing of SAGE II transmission profiles, is to determine the tangent altitude (altitude registration) and the point on the Sun that it intersects with (Damadeo et al., 2013). Once transmission profiles are obtained, constituent profiles including ozone are produced using a straightforward linear species separation algorithm that relates aerosol extinction at 452, 525 and 1020 nm to an unknown aerosol contribution at 600 nm where ozone is effectively inferred. SAGE II ozone values are reported whenever they can be produced by the processing software. The uncertainties reported for ozone observations are generally reliable assessments of ozone data quality. The altitude range reported for data products is limited to the range where relevant observations are made over the lifetime of the instrument. For instance, ozone can be reported from 0.5 to 70 km while aerosol extinction coefficient is reported no higher than 40 km. Since the ozone contribution at 600 nm is usually much larger than that by aerosol, the retrieval of ozone number density is robust and produces high quality data over a broad range of altitude and conditions. However, there are well know exceptions to this generality and, like any data set, SAGE II ozone measurements require cautious use. For instance, given that aerosol extinction coefficient at 600 nm is interpolated across a broad range in wavelengths (a factor of almost 2), it is hardly surprising that the interpolation is not perfect and, when aerosol extinction coefficient is high, the potential for introducing artifacts in the ozone data set is significant. While enhanced aerosol loading in the atmosphere is not common during the SAGE II mission, the Mt Pinatubo eruption of June 1991 increased aerosol extinction coefficient substantially (primarily below 25 km) by as much as a factor of 1000. SAGE II ozone data quality in this extreme period is clearly negatively impacted (Yue et al., 1995). Furthermore, the presence of clouds, either tropospheric or polar stratospheric, and occasional smaller volcanic eruptions can have a similar deleterious impact on ozone data quality and therefore need to be accounted for before SAGE II data are used.





## 3 Current data usage rules for SAGE II ozone

SAGE II measurements are reported during periods of high aerosol loading, i.e. periods where aerosol extinction coefficients are large and many current and past data usage rules have been designed to remove anomalous water vapour and ozone observations that are associated with these periods, the presence of clouds or instrument-related artifacts (e.g. Wang et al., 2002). No recent review has been done on whether or not the current screening recommendations of the SAGE II ozone data are still applicable to the most recent version of the SAGE II data, resulting in potentially good data points being removed

from the data set or bad data points remaining, which in turn can lead to biases in e.g. trends that are derived from the SAGE II observations.

Here we will discuss data usage rules that have been applied in screening SAGE II ozone data before their use in the generation of homogeneous long-term ozone data sets as described in Froidevaux et al. (2015); Hassler et al. (2008, 2018) and Davis et al. (2016). The data rules outlined below were derived by others and are the most predominately rules found in the

literature and are commonly applied usage rules. We have not included every proposed filter nor should those given below be considered, in part or in total, as a canonical set but they are rather simply a collection of commonly used data usage rules for SAGE II ozone. The rules that are further looked at in this study (hereafter referred to as 'current rules') are:

1. Exclude all values between 23.06.1993 and 11.04.1994 between 15 km and 50 km if the error is bigger than 10 % (Hassler et al., 2008).

2. Wang et al. (2002) suggested to remove all data if error is greater than 300 %. This rule was later adapted to only exclude ozone values above 35 km if error is bigger than 300 % (Froidevaux et al. (2015); Davis et al. (2016); Hassler et al. (2018), SAGE II v7 release notes).

3. Exclude all ozone values below or at 35 km if error is bigger than 200 % (Davis et al. (2016); Hassler et al. (2018), SAGE II v7 release notes)

4. Exclude ozone values below or at the level where 525 nm extinction $> 1\times10^{-3}$ km$^{-1}$ and the extinction ratio 525 nm/1020 nm is < 1.4 (SAGE II v7 release notes, Froidevaux et al. (2015); Davis et al. (2016); Hassler et al. (2018)).

5. Exclude all values between 30 and 50 km where the uncertainty is > 10 % (Froidevaux et al. (2015); Davis et al. (2016); Hassler et al. (2018) and SAGE II v7 release notes).

6. Exclusion of all data points at altitude and below the occurrence of an aerosol extinction (525 nm) value of greater than

$6\times10^{-3}$ km$^{-1}$ (Davis et al. (2016); Hassler et al. (2018) and SAGE II v7 release notes).

7. Eliminate all data below 23 km between July 1991 and December 1993 for excessive aerosol (Hassler et al., 2008, 2018).

8. Exclude all values between 10.5 and 24.5 km if ozone > 10 ppm. This rule removes large ozone values and was developed based purely on visual inspection (Rind et al., 2005; Hassler et al., 2008).





9. Exclude all values above 25 km if ozone > 100 ppm. This rule removes large ozone values and was developed based purely on visual inspection (Rind et al., 2005; Hassler et al., 2008).

10. Exclude all value at pressure < 3 hPa if ozone > 50 ppm. This rule removes large ozone values and was developed based purely on visual inspection (Rind et al., 2005; Hassler et al., 2008).

11. Outlier screening by removing all values that are more than $10\sigma$ away from the monthly mean value for a given latitude band ($15°$ zones), longitude ($90°$ quadrants) and altitude (0.5 km grid) (Rind et al., 2005; Hassler et al., 2008)). This rule was later adapted to remove values that are farther than $3\sigma$ away from the mean in $10°$ latitude bins (Davis et al., 2016; Hassler et al., 2018).

Up to seven out of the 11 recommendations were used in the studies by Froidevaux et al. (2015); Hassler et al. (2008, 2018) and Davis et al. (2016), rules 1, and 8 to 10 were included here as they have been used in earlier versions of the vertically resolved ozone database described in Hassler et al. (2008). These rules seem to have been replaced by rule 11, the outlier screening rule in more recent publications. However, Froidevaux et al. (2015) for example, did not apply any outlier screening rule before using SAGE II ozone data. There seem to be some confusion about whether or not to apply outlier screening rules and which rule should be applied to SAGE II ozone. Here, we will provide a new outlier screening recommendation that is applicable to not only normally distributed data but also skewed data sets, and which does not rely on visual inspection of the data set which is a rather subjective screening method.

## 3.1 Comments on the current data usage rules

It is the purpose of any data usage rules to differentiate between usable data, including high quality, atypical, or noisy but unbiased data, and unusable, or best to avoid data, including data where a bias is likely to be an issue and data with excessively high noise levels suggesting that they contain no useful information. There is, in principle, no need for any data usage rules to be applied, if the data are well behaved in the sense that they only consist of random noise with zero mean. In fact, for an ideal data set, it would be incorrect to follow common practice and eliminate data based on relative error because low values, that represent the negative tail of a well-behaved distribution of noisy data, are preferentially eliminated. As a result, the mean of the data set can be biased high even when the family of data points is itself unbiased. An example of this effect can be seen in Fig. 2, where the application of current rule #3 leads to unexpected results. In this example, measurement noise is relatively large and spreads the reported ozone values, at that altitude and latitude bin, across the zero line. Using the relative errors to flag data as outliers, leads to removal of ozone values lying in a band centered at zero, while data larger in magnitude of both signs will remain in the data set. While this is an example of unintended consequences, it is a good example of conditions in which an argument could be made to eliminate no data and averaging allowed to do its work. Removing all points less than a fixed relative error will bias the results (Fig. 2) to a much larger positive value. However, this issue could be mitigated by using a relative error rule in which the ozone uncertainty in number density is compared to the average value of ozone within the averaging window rather than individual measurements so that it becomes an assessment of the size of the reported uncertainty rather the size of reported ozone number density.

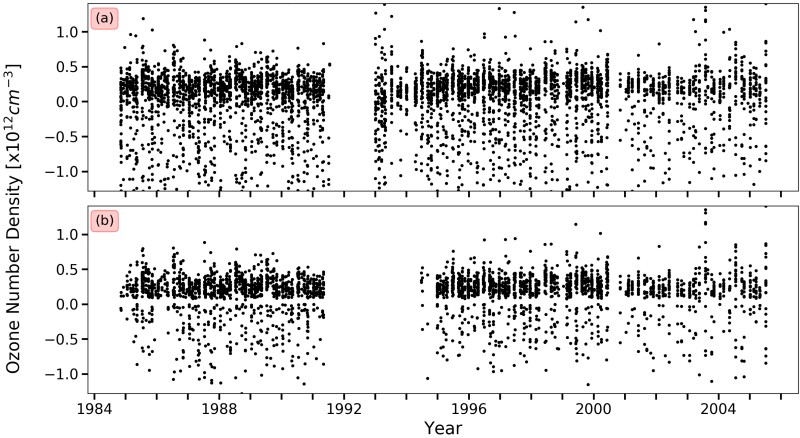

**Figure 2.** Ozone number density at 16 km between 5° N and 15° N from SAGE II, (a) all ozone data included in the SAGE II data in that latitude bin, (b) remaining ozone data after current rule #3 has been applied. Note that the y-axes have been fixed to the [-2,2] range to illustrate the removal of ozone data close to the zero line.

The usage rules outlined above were applied to SAGE II ozone data located at a certain altitude and within a certain 10° latitude band. To investigate the impact of each rule on the SAGE II ozone data, all rules were applied individually to the whole data set at a given altitude and within a given latitude band. Therefore, the order in which the rules are applied does not impact

which data points are flagged for removal and data points might be flagged more than once. SAGE II ozone observations at 20 km and 16.5 km altitude and between 5° S and 5° N and 25° S and 15° S throughout the duration of the SAGE II mission are shown in Fig. 3 together with the flagged ozone values that would be removed by applying the current rules as indicated in the legend. Note that some of the data usage rules only apply to higher altitude ranges, and therefore no data are removed at the given altitudes. The percentage of the ozone values that remain and that would be removed by applying the rules is shown in

the legend; note that the sum of all percentages will be greater than 100 % as the current rules are applied individually to the data set, i.e. the data point can be removed more than once.

At 20 km, overall 10 % of the ozone data are removed by applying the current rules outlined in Sect. 3, and rule #4 and rule #6 remove the majority of the ozone data due to aerosol extinction values exceeding a threshold. However, is it clear that applying the current rules to the SAGE II ozone data set are too restrictive by removing ozone measurements that appear

unaffected by aerosol especially during the period of the Nyamuragira/Nevado del Ruiz eruptions (1985/1986), see Fig. 3a. At lower altitudes (16.5 km), 15 % of the ozone data are removed when applying the data screening rules, with the majority being removed due to rule #4, which again is based on aerosol extinction values. While aerosol extinction values are enhanced due to volcanic eruptions, this should not equal bad ozone data as it can be seen in Fig. 3b. Here, ozone values just before the

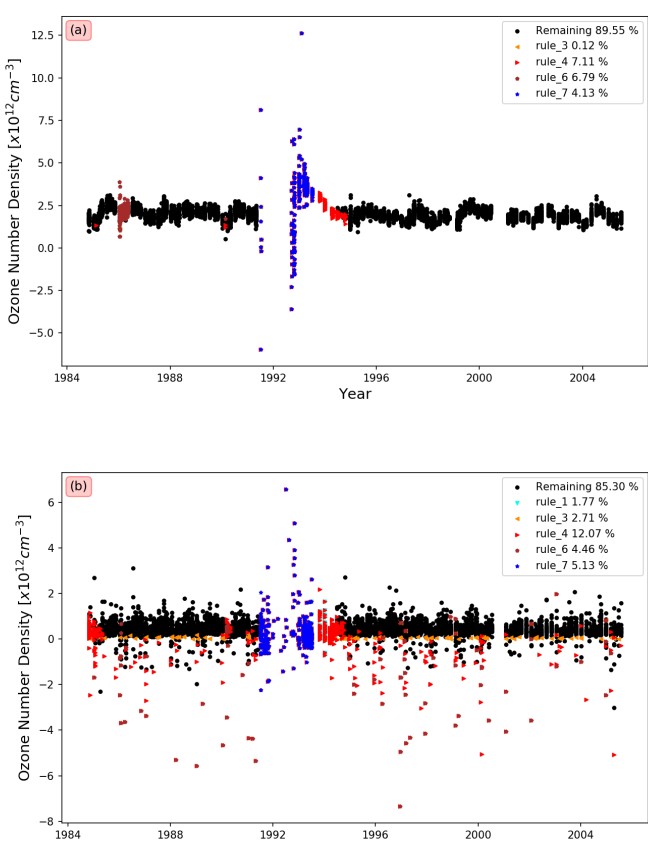

**Figure 3.** SAGE II ozone number density at 20 km between 5° S and 5° N (a) and 16.5 km between 25°S and 15°S (b) before and after data usage rules (Sect. 3) have been applied as indicated in the legend. Each rule is associated with a percentage of data points that will be removed from the data sets, if the rule is applied. For more details see text.

Nyamuragira/Nevado del Ruiz eruptions and at the end of the Mt Pinatubo eruption period (1994/1995) seem as valid as any
other ozone data point and should not necessarily being removed. Furthermore, applying rule #3 leads to ozone values around zero to be removed, which as discussed above, biasing any mean values calculated from this set of data. Overall the data usage recommendation as described above appear unnecessarily restrictive and relying on current aerosol extinction coefficient rules to capture the impact of aerosol on the quality of the ozone data is not advisable. This result confirms the finding of a recent study by Damadeo et al. (2018), who concluded that the Wang et al. (2002) filtering recommendations are overly conservative
and need to be revisited.

While Hassler et al. (2018) did not directly employ a cloud presence filter as a usage rule, Rind et al. (2005); Hassler et al. (2008); Davis et al. (2016) have used the SAGE II cloud presence flags to eliminate data when the presence of clouds has been inferred. Applying cloud flags to SAGE II data may be justified by the inhomogeneity of clouds as they are present in SAGE-





like observations (Thomason and Vernier, 2013), as opposed to strictly confining the usage rules to the magnitude of aerosol
extinction. However, it is not clear that the impact on ozone by enhanced extinction from cloud presence can be significantly
different from that caused by enhanced aerosol, which has its own associated usage rules. Since the detection of cloud presence
in SAGE II observations is ambiguous at best (Thomason and Vernier, 2013) any cloud presence rule has been excluded in this
study. While, further in depth investigation of the cloud effect on ozone data quality may be worthwhile, it is beyond the scope
of this study.

In addition to the 11 screening rules listed in Sect. 3, Froidevaux et al. (2015) and Davis et al. (2016) removed all SAGE
II ozone profiles associated with 'short events' during the 1993 and 1994 period as described in Taha et al. (2004). The 'short
events' rule is based on the beta angle which is a spacecraft/event characteristic, i.e. the beta angle is the elevation angle of the
Sun with respect to the orbital plane of the spacecraft. Most significantly, beta angle governs the duration of an event where
the larger the magnitude, the longer an event lasts. Increasing event duration corresponds to expanded spatial extent, i.e. for
any event, the latitude/longitude of tangent altitudes moves at roughly 7 km s$^{-1}$, which, in turn, reduces the applicability of the
assumption of spherical homogeneity of the atmosphere used by the retrieval algorithm. The 'short events' period relates to the
time when SAGE II data was collected during an event that was shortened to reduce power requirements following the failure
of a series of cells in the spacecraft battery. Taha et al. (2004) found that sunrise events with beta angle between -47° and +47°
and sunset events with beta angle less than -45° and greater than +45° are the most affected events and need to be removed.
Data for these events are notably noisier than unaffected events (Damadeo et al., 2013). Short events were a part of SAGE II
version 6.2 but are excluded from the processing within the retrieval in version 7.0 (Damadeo et al., 2013) and therefore the
rationale behind this rule is less compelling and this rule should be eliminated and not be applied to SAGE II v7 ozone data.
Therefore, this rule is excluded from the list of 'current data usage rules' listed in Sect. 3.

## 4   How measurements are made and how that should impact usage rules

In revising the SAGE II ozone data usage rules, the primary goal is to simplify them and to retain as much data as possible
without compromising the data quality of the remaining data. A guiding principle for the development of new usage rules is
that any information needed in the revised rules must be readily accessible to other users and, hence, reproducible by other
users of the data set. This excludes for instance the use of SAGE II transmission data since it is not routinely made available
to users. This is important to note since one goal of the way the new usage rules are devised is to make them reflect the way
in which the measurements were made. Therefore, as outlined below, the line-of-sight (LOS) values for aerosol extinction
coefficient will be used. Since this is not routinely available in the data set, we calculate the LOS aerosol optical depth using
the reported the aerosol extinction profiles integrated along the LOS path that is approximated using a simple geometric
model, thereby neglecting the impact of refraction. In addition, aerosol extinction at 600 nm is estimated within the SAGE II
retrieval algorithm but it is not included in the data product. Therefore, this parameter is estimated by using a simple Ångström
coefficient approximation based on extinction reported at 525 and 1020 nm as described below. The approximations outlined





below are adequate for the application of developing new data usage rules and are readily reproducible by any user of the SAGE II ozone data.

Neglecting small contributions by nitrogen dioxide and water vapour, the SAGE II line-of-sight optical depth $t$ at 600 nm is given by:

$$\tau(z_0) = \int_{z_0}^{\infty} (\sigma_{O3} n_{O3}(z) + \sigma_m n_m(z) + k_a(z)) \frac{dl(z)}{dz} dz \tag{1}$$

where $z_0$ is the tangent altitude, $\sigma_m$ is the molecular cross section, $n_m$ is the neutral density number density profile, $\sigma_{O3}$ is the ozone absorption coefficient, $n_{O3}(z)$ is the ozone number density, and $k_a$ is the aerosol extinction coefficient profile at 600 nm. The derivative $dl(z)dz^{-1}$ is the distance travelled along the line of sight per unit change in altitude. Values for $n_{O3}$ and $n_m$ are available directly from the SAGE II data product files, and $dl(z)dz^{-1}$, neglecting the effects of refraction, can be

computed from simple spherical geometric considerations. In discrete form, $\Delta l$, that reflects the 0.5 km vertical spacing of SAGE II reporting altitudes, is computed as:

$$\Delta l_{i,j} = 2 \times \left( \left[ (R + z_j)^2 - (R + z_i)^2 \right]^{\frac{1}{2}} - \left[ (R + z_{j-1})^2 - (R + z_i)^2 \right]^{\frac{1}{2}} \right) \tag{2}$$

where $i$ is the LOS optical depth altitude and $j$ are the data levels above the measurement (so $j > i$). The value $\Delta l_{i,i+1}$ is applied to the data at level $i$.

From Eq. 1 it is clear that the measured LOS optical depth at any altitude is dependent not just on components at that altitude but also at all levels above it. A strength of the occultation measurement is that given the limb-viewing geometry the contribution is dominated by the lowest few altitude levels. Figure 4a shows the value of $\Delta l_{i,j}$ as a function of altitude and a number of tangent altitudes. While these weighting functions are heavily weighted toward the tangent altitude, the shape of the constituent profile (e.g. ozone) can significantly alter the altitudes that dominate a measurement. Figure 4b shows an ozone

profile for June 1999 between 5° S and 5° N. In this example, the ozone profile has a broad peak around 25 km and decreases rapidly below the peak. The weighting of ozone measurements as a function of altitude are shown in Fig. 4c. It is the product of Fig. 4a and Fig. 4b using only the ozone portion in the integral of Eq. 1. For measurements with tangent altitudes at and above above the ozone peak (around 25 km), the measurement is heavily weighted to the tangent altitude. However, for tangent altitudes below the peak, measurements are more heavily affected by the overlying ozone, such that the total line-of-sight ozone

amount primarily consists of ozone several kilometers above the tangent altitude.

Overall, the fraction of the total LOS optical depth due to ozone at the measurement altitude can be estimated as

$$f = \frac{\sigma_{O_3} \times n_{O_3}(z) \times \Delta l_{i,i+1}}{\tau_{LOS}(z)} \tag{3}$$

where $\sigma_{O3}$ is the ozone absorption coefficient , $n_{O3}(z)$ is the ozone number density at altitude $z$, $\Delta l_{i,i+1}$ is the length of the LOS path through the altitude bin $i$ with the tangent point being at altitude z, and $\tau_{LOS}(z)$ is the LOS optical depth at altitude

$z$ calculated using Eq. 1. Measurement fractions, $f$, for a given latitude band (5° S and 5° N) from 10 to 70 km are shown in Fig. 5. Measurement fractions maximize between about 25 and 50 km between 20 and 30 %, which is quite consistent from



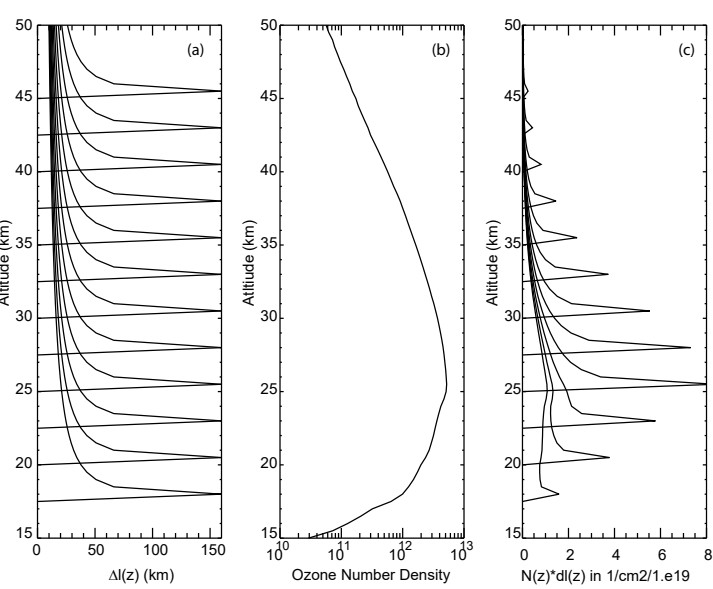

**Figure 4.** (a) Profiles of $\Delta l_{i,j}$ (in km) for tangent altitudes from 18 to 45.5 km in 2.5 km steps in tangent altitude, (b) a representative ozone profile $n_{O3}(z)$ in $cm^{-3}$ , and (c) the product of the curves in (a) with the ozone profile shown in (b) so $\Delta l_{i,j} \times n_{O3}(z)$ in $cm^{-2} \times 10^{19}$.

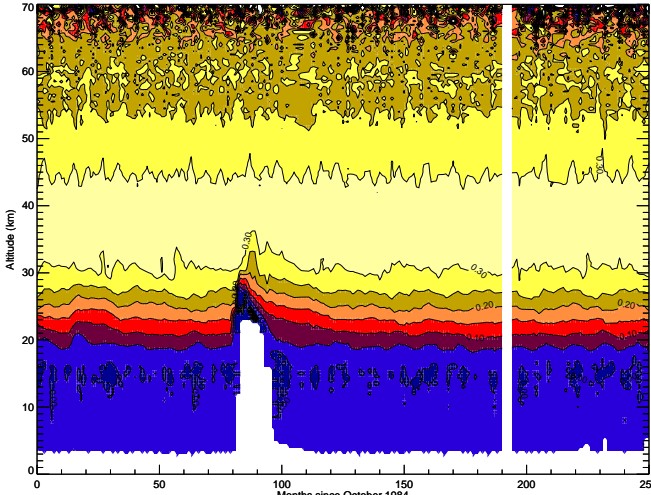

**Figure 5.** Measurement fraction, $f$, for 5° N for all SAGE II data (no clearing) for the entire lifetime of the instrument. Note that the Mt Pinatubo period, particularly from month 81 (June 1991) to through roughly month 100 (the end of 1992) show clear reduction of the value of $f$ above 18 km to as high as 35 km. Though several other volcanic events occur during the SAGE II lifetime (e.g. Ruiz/Nimuragyra), none of these are readily apparent in $f$ since they were an order of magnitude or more smaller than the Mt Pinatubo event.

year to year except during the 1991-1994 period, where the stratosphere is strongly impacted by the June 1991 eruption of Mt Pinatubo. In this latitude band (tropics) the ozone fraction of the optical depth in the Upper Troposphere/Lower Stratosphere (UTLS) is only a few percent of the total LOS optical depth. As a result, performing ozone measurements from SAGE II are
extremely challenging at these altitudes and they are very sensitive to even modestly enhanced aerosol.

Ozone measurement uncertainty is usually dominated by the estimates of transmission uncertainty that are carried through the retrieval process to ozone and other product uncertainties. While transmission uncertainty varies slowly with altitude, the rapid decrease in ozone fraction below the ozone peak results in ozone uncertainties due to transmission uncertainty to increase rapidly. Overall, transmission uncertainty is an unbiased source of error and roughly Gaussian so it can be reduced
by averaging multiple measurements as is done in Hassler et al. (2018) and other ozone climatologies. Of course, at very high altitudes (>60 km) and low altitudes (at and below the tropopause), the uncertainty can become sufficiently large that a meaningful measurements can no longer be made and no amount of averaging will produce a reliable result.

Other than ozone itself, there are two additional components to the LOS optical depth at 600 nm; (i) molecular scattering, which is significant in the lower stratosphere and above 45 km and (ii) aerosol extinction, which is highly variable and the
most likely contributor to biases in the measurements. The molecular scattering correction is generally small and only slowly varies in an altitude/latitude bin and is therefore not of particular importance for the following discussion. The second term, however, is particularly noteworthy during the Mt Pinatubo period, but it can also be episodically significant for smaller eruptions such as the Nyamuragira/Nevado del Ruiz (1985/1986) eruptions and when clouds or polar stratospheric clouds are





present. As previously noted, aerosol extinction at 600 nm must be inferred from aerosol extinction measurements at 452, 525
and 1020 nm. Aerosol correction usually contributes to the measurement uncertainty in the stratosphere, as ozone dominates
aerosol extinction at 600 nm. However in the upper troposphere, where clouds are present, or during significant volcanic
eruptions, even small deficiencies in the estimation process of aerosol extinction at 600 nm can cause significant deviations
in the retrieved ozone values. The version 7.0 algorithm that is used to derive SAGE II ozone data, includes an estimate of
the possible magnitude of bias introduced in the retrieved ozone values due to the estimation of aerosol extinction at 600 nm.
However, this estimate itself is highly uncertain and therefore its use in studies and efforts in e.g. generating a merged ozone
database such as Hassler et al. (2018); Davis et al. (2016) is limited.

In light of this, a number of ozone data usage rules based on aerosol extinction coefficient apart from provided uncertainties
have been proposed by Wang et al. (2002) and have been used in Hassler et al. (2018) and Davis et al. (2016). These include
rules that exclude much of the ozone in the lower stratosphere following the Mt Pinatubo eruptions (current rules #1 and #7).
Like the importance of the ozone over burden to ozone uncertainty at an altitude, a similar and somewhat hidden aspect of
ozone measurement uncertainty is its sensitivity to enhanced aerosol (or cloud) above the measurement altitude even though
aerosol at that altitude is well within nominal levels. This sensitivity is not reflected in any of the current SAGE II ozone data
usage rules and accounting for this possibility is a goal for this study.

Nominally, accounting for the burden of material above the tangent altitude should be straightforward as every SAGE II
event has an intermediate product of transmission profiles at the wavelengths available from the instrument. However, this
product is not routinely made available from what is essentially a heritage mission. Developing generally usable rules requires
us to use only publicly available products. Fortunately, since the rules will be applied in only a semi-quantitative manner, we
can simulate the LOS values in a straightforward way using products available in the primary data product files. In addition,
we simplify the overburden test to consider only aerosol, since molecular and ozone contribution do not vary a great deal over
the lifetime of the instrument. For our aerosol test, we simulate 600 nm aerosol extinction coefficient using the aerosol portion
of Eq. 1, the path length from Eq. 2 and a reconstructed aerosol extinction coefficient at 600 nm. While aerosol extinction
coefficient at 600 nm is computed during data processing, it is never explicitly examined or retained as a data or Q/A product.
For this study, we estimate extinction at 600 nm using SAGE II aerosol extinction coefficient at 525 and 1020 nm using a
simple Ångström coefficient, $\alpha$ approach where:

$$\alpha = -\frac{\ln \frac{k_{a_{525}}}{k_{a_{1020}}}}{\ln \frac{525}{1020}} \qquad\qquad k_{a_{600}} = k_{a_{1020}}(\frac{600}{1020})^{-\alpha}$$

Aerosol extinction coefficient is only reported to 40 km and very small negative values above 30 km are regularly reported in the data files. For the purposes of this study, at any time when either aerosol extinction at 525 or 1020 nm is reported as negative, the aerosol extinction coefficient at 600 nm is set to zero. Generally, $k_{a_{600}}$ is between $k_{a_{525}}$ and $k_{a_{1020}}$ (closer to $k_{a_{525}}$) and $\alpha$ is usually between 0 (clouds and high amounts of volcanic aerosol) and 3 (low amounts of aerosol).

The relative contributions by ozone, aerosol, nitrogen dioxide, water, and molecular scattering vary by season, the phase
of the Quasi-Biennial Oscillation (QBO), latitude, and altitude. Furthermore, these contributions are modulated by volcanic
eruptions, which is always ignored in the construction of ozone usage rules under the assumption that the effects are small or





invariant enough for the purposes of identifying outliers. While this assumption is probably true, it may be worth taking a look into in a future study.

## 5    Development of new ozone data usage rules

In this section, the rationale behind the development of new SAGE II ozone data usage rules is described. The goal was to reduce the data screening recommendation to as few as possible and to consider two new recommendations, where one is based on LOS aerosol optical depth; and the other is a basic outlier identification rule. Additional recommendations were only considered when clear deficiencies were uncovered following implementation of the primary rules.

### 5.1    LOS aerosol optical depth and ozone data quality

It is well known that SAGE II ozone data quality is modulated by variations in aerosol extinction coefficient (e.g. Steele and Turco, 1997; Wang et al., 2002). This is primarily due to the relatively sparse spectral sampling of the instrument and the need to effectively infer aerosol extinction at 600 nm from measurements at 525 and 1020 nm (see Sect. 4). The strength of the ozone signal in the lower stratosphere is generally more than adequate for a robust retrieval of ozone concentration. However, enhanced aerosol from clouds and particularly from the major Mt Pinatubo eruption in 1991, can reverse the weight of observations at 600 nm from ozone toward aerosol. As shown in Fig. 5, even in the main ozone layer in the 20 to 30 km range, the fraction of the signal from ozone at the measurement altitude drops by factors of 3 or more during the Mt Pinatubo period. Therefore it is not surprising that the quality of the ozone measurements declines under these circumstances.

Figure 6 shows an extinction coefficient profile (at 1020 nm), as retrieved from the measured transmissions, that encounters a dense aerosol layer, in this case probably a cloud, at about 15 km. While the extinction profile down to this altitude remains robust, there are substantial oscillation in the profile below the top of the aerosol layer. At the same time the LOS aerosol optical depth remains well behaved below this altitude (Fig. 6). The oscillations reflect numerical instability, resulting from very large factor variations (up to 1000) in extinction coefficients over a narrow altitude range and such features invariably do conveniently occur exactly within the SAGE II altitude grid. While the reported uncertainties on the SAGE II data show that this data is of poor quality, solely relying on these extinction values to reflect the impact of aerosol on the quality of the ozone data is not advisable and other measures on how best to flag ozone data that are affected by enhanced aerosol are required.

The dependence of ozone on aerosol LOS optical depth, derived from the aerosol extinction coefficient at 600 nm (see Sect. 4), for a pair of altitude/latitude bands is shown in Fig. 7. There is a cluster of observations at low aerosol LOS optical depth and then a long tail toward higher values mostly occurring during the multi-year recovery from the Mt Pinatubo eruption through the middle 1990s. For much of the aerosol LOS optical depth domain, ozone shows no clear variation with increasing optical depth but at values exceeding about 3 (vertical line in Fig. 7), ozone appears to decline. While an aerosol-related ozone change cannot be excluded, this decline is generally taken as evidence of an aerosol artifact in the ozone measurements. Examining all latitude/altitude bands we observe that the impact of aerosol on ozone does not occur for aerosol LOS optical depths below 3 and in some situations does not appear until values exceed 4. Since the volume of data with LOS optical depths between 3 and

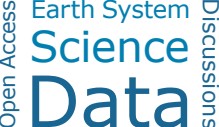

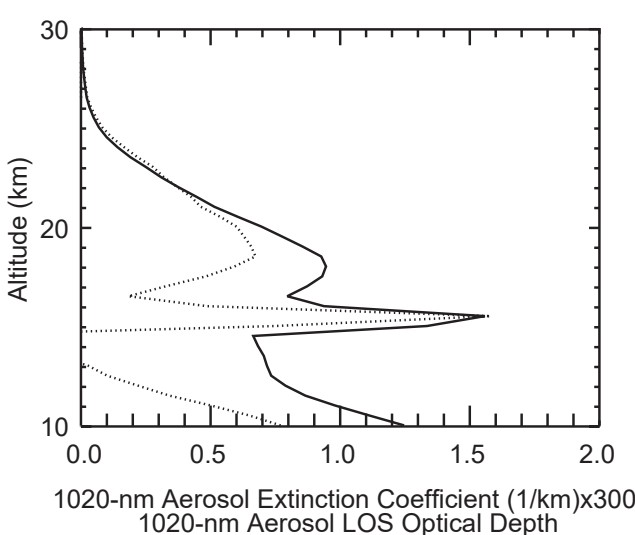

**Figure 6.** Single profile of 1020-nm aerosol extinction coefficient x 300 (dashed line) along with the computed LOS optical depth (solid line). The extinction coefficient, particularly below the cloud located near 15 km is clearly not representative of the impact of aerosol on ozone determinations at that altitude.

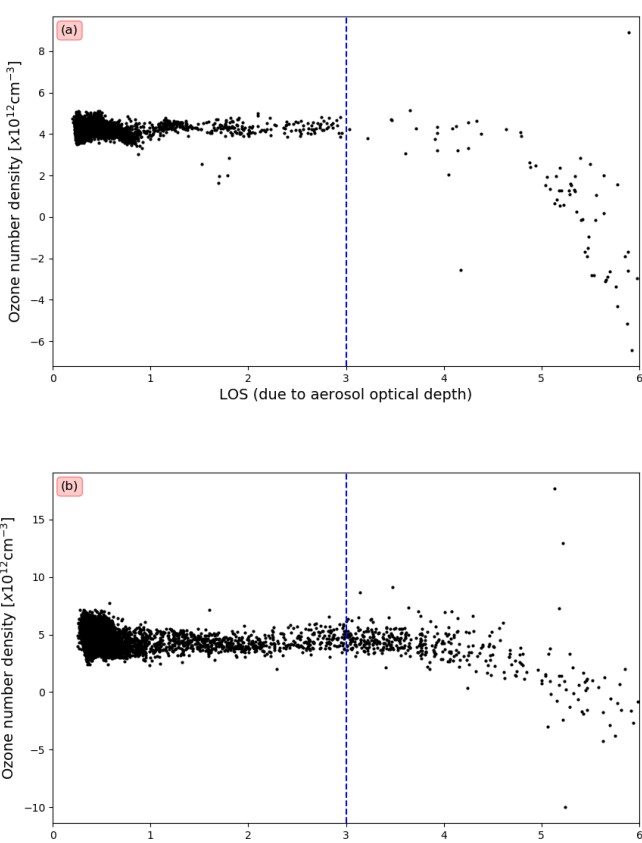

**Figure 7.** Ozone number density versus LOS optical depth due to aerosol at (a) 24 km between 5 and 15° S and (b) 21 km between 35 and 45° N.

4 is small, we conservatively use 3 as the cut off value at all latitudes bands and altitudes, i.e. our recommendation is to remove all ozone measurements at any given altitude and within a given latitude band if the corresponding aerosol LOS optical depth exceeds a value of 3.

    After the new aerosol rule is applied to the SAGE II ozone data set (hereafter referred to as 'LOS optical depth rule'), the biggest outliers caused by, e.g. aerosol enhancements in the atmosphere, are removed. SAGE II ozone at 16 km and 21 km

altitude between 5° and 15° S for each month of the observing period is shown in Fig. 8. The blue dots in Fig. 8 indicate the ozone data that will be removed due to the aerosol LOS optical depth rule. At 21 km, this rule mostly removes ozone data during the Mt Pinatubo eruption period, which lead to high ozone concentrations to be retrieved, and even negative ozone concentrations in some months. Outside the Mt Pinatubo period, the ozone values remain un-affected by this screening recommendations (Fig. 8 a). At 16 km, the LOS increases as to be expected, and the LOS optical depth rule removes a decent

amount of data, especially the negative ozone concentrations that were retrieved and are provided with the SAGE II data.



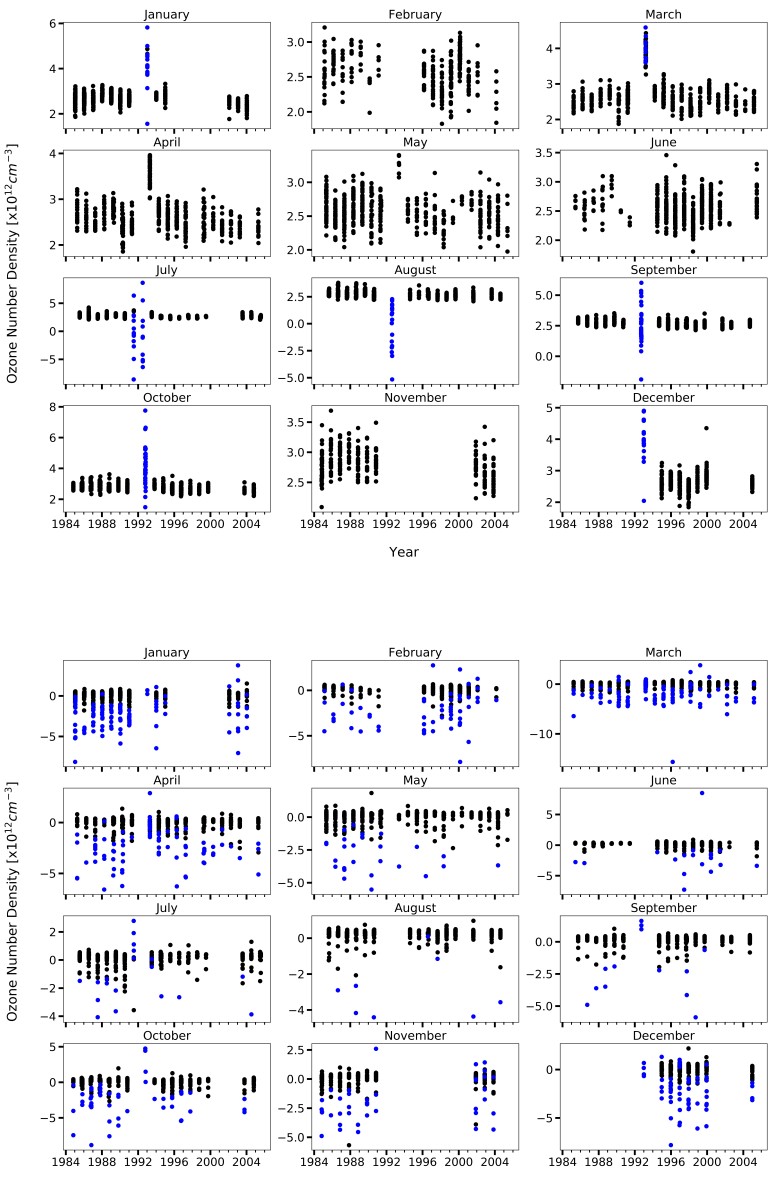

**Figure 8.** Remaining SAGE II ozone at 21 km (top) and 16 km (bottom) between 5° and 15° S for each month of all years (black) after the LOS optical depth rule has been applied to the ozone data set (blue dots).





## 5.2 Usage rule according to reported relative uncertainties

Applying the LOS optical depth rule to the SAGE II ozone data set did not remove all deficiencies from the data set. Looking into the remaining features in more detail, we discovered that there are some ozone values, where the relative uncertainty was set to exactly 200 %, which to us, seemed surprising. Further investigation lead us to the following explanation for these

uncertainties and how they should be treated:

As the version 7.0 data product was developed, it was noted that, under some circumstances, the calculated uncertainty for ozone concentration was much too small when negative values were reported. During investigations for the development of new ozone data usage rules, it was observed that the defect occurs primarily in the lower stratosphere and troposphere where the aerosol LOS optical depth exceeds 2 but, for reasons not immediately clear, this does not affect all data where this aerosol level

of 2 is exceeded. Mostly likely this reflects a failure in the empirical means of accounting for the deleterious impact of excess aerosol LOS optical depth in the ozone retrieval (Damadeo et al., 2013). The fixed 200 % uncertainty value is intended as a means to differentiate between negative retrieved values with high uncertainties and retrieved values with low but unrealistic uncertainties. Negative retrieved ozone values and their associated uncertainties remain intact at higher altitudes, as this is more often a result of noise in the data rather than a deleterious effect on the retrieval algorithm from high aerosol extinctions or

clouds. So since we are certain that the 200 % uncertainty data is the result of an aerosol issue but no single value of aerosol LOS optical depth can be identified, we have implemented a rule that removes all ozone observations where the uncertainty is reported as exactly 200 %. We recommend to apply this rule first, before any other rule is considered.

The effect of applying 200 % rule on the ozone data is illustrated in Fig. 9, showing SAGE II ozone observations at 12 km between 65° S to 75° S latitude band. In this case, substantial scatter of ozone observations is apparent. There are some

high values that occur during the Mt Pinatubo period, which may reflect aerosol contamination of the ozone product plus data points stretching downward into negative values throughout the SAGE II lifetime. Blue dots in Fig. 9a represent ozone data points with uncertainties reported as exactly 200 % and data that are eliminated from the data set; essentially all the negative values have been eliminated. Figure 9b shows the result for eliminating all data where the aerosol LOS optical depth exceeds 2 (marked as blue dots). In this case, essentially all of the points that are eliminated by the 200 % rule are also eliminated,

as well as those point with very high ozone values during the Mt Pinatubo period. However, it is also clear from Fig. 9b that some points that fall well within acceptable bounds are also being removed and overall 12.1 % of the data are removed in that latitude bin. This problem is ubiquitous in the lower stratosphere and it is clear that simply eliminating all data with LOS aerosol optical depth exceeding a value of 2 is not acceptable; compare 0.8 % of the data that are removed by applying the 200 % rule with 12.1 % of the data that are removed by using the LOS aerosol optical depth exceeding a value of 2 criteria.

## 5.3 Statistical outlier identification

After applying the new LOS optical depth rule outlined above, the distribution of ozone values within latitude/altitude bins is dominated by geophysical variability and unbiased measurement noise. This generally results in distributions that are roughly Gaussian in shape and applying noise filtering such as current rule #11 (cf. Sect. 3), removing ozone values that are 3$\sigma$ away

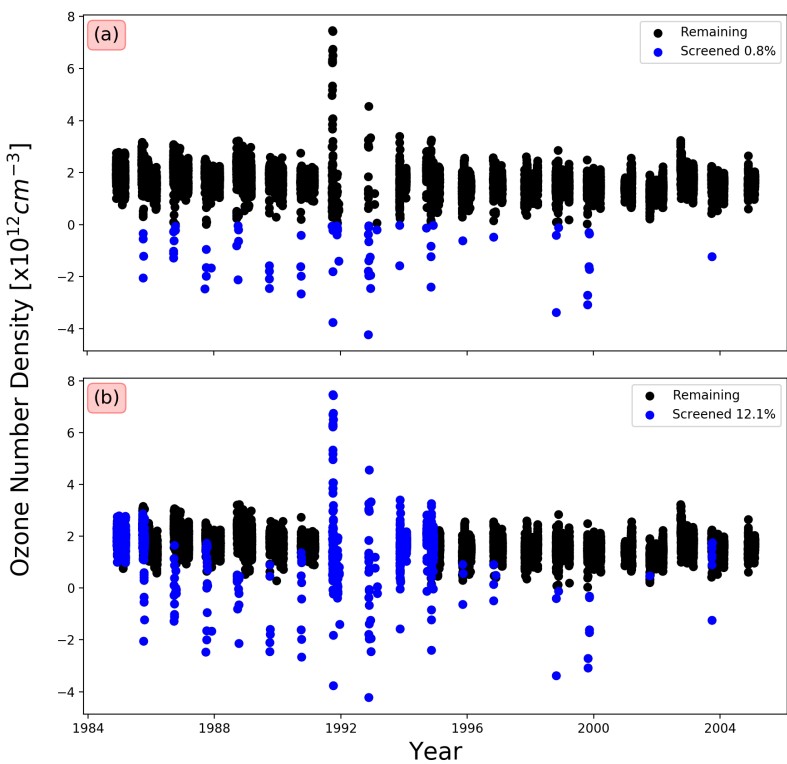

**Figure 9.** Ozone number density at 12 km between 65 and 75° S. (a) SAGE II ozone data at the given latitude bin and altitude that remain (black dots) after ozone values with a relative error of 200 % are removed (blue dots), (b) SAGE II ozone data at the given latitude bin and altitude that remain (black dots) after ozone values where the aerosol LOS optical depth exceeds 2 were removed (blue dots).

from the mean of the distribution of ozone values, can capture most obvious outliers. However, we note that there are many
situations in which the distribution of ozone within the latitude/altitude bins demonstrate a clear skewness in their shape (e.g., Fig. 10). The skewness is the result of strong variations in ozone concentration with spatial coordinates like equivalent latitude or potential vorticity (PV) that do not exactly coincide with latitude (Schoeberl et al., 1993). For instance, strong variations in ozone across the polar vortex boundary that is asymetrically situated across the pole can lead skewness in the distribution of ozone in a latitude bin. Similarly, the ozone distribution in tropical latitudes can be created by gradients of ozone across a
meandering tropical pipe (see Fig. 10). Therefore, we find that applying a simple Gaussian filter tends to preferentially remove ozone values on the broad tail of the skewed distribution with a concomitant impact on statistical values, particularly means, for these bins. We considered performing the outlier test in equivalent latitude space as equivalent latitude is based on PV and is a widely used diagnostic for isentropic transport in the stratosphere and upper troposphere. While we found that the skewness





of the ozone distributions observed in equivalent latitude space is reduced compared to using geographical latitude space (not
shown), significant skewness remained and an alternative approach was developed.

To mitigate the skewness issue in outlier detection, we employed a skewed distribution outlier test as the third and last rule for
SAGE II ozone filtering. Here, we apply a method developed by Hubert and Van der Veeken (2008) to detect outliers which does
not need the assumption of symmetry nor rely on visual inspection. It is based on the quartile text in which outliers for a uni-
variate (continuous, unimodal) data set, $X_n = x_1, x_2, ..., x_n$ are inferred outside of the bounds of [Q1-1.5×IQR,Q3+1.5×IQR],
where Q1 is the first quartile for the distribution X, Q3 is the third quartile and IQR is the interquartile range, defined as $Q_3 - Q_1$.
While, for data coming from a normal distribution, the probability to lie beyond the whiskers is approximately 0.7%, this per-
centage can be much higher if the data are skewed . For skewed data, Hubert and Van der Veeken (2008) employ the statistical
quantity medcouple to modify the outlier bounds to better encompass the distribution of X. The medcouple ($MC$) is a robust
measure of skewness of a distribution and is bound between a value of -1 and 1. For a symmetric distributions, $MC$ is zero,
for left- and right-skewed data the medcouple is negative and positive, respectively. The medcouple is defined as the median of
the kernel function ($MC(X_n) = med\ h(x_i, x_j)$) where $h(x_i, x_j)$ is defined as:

$$h(x_i, x_j) = \frac{(x_j - med(x)) - (med(x) - x_i)}{x_j - x_i} \qquad (4)$$

evaluated over all couples $(x_i, x_j)$ where $x_i$ is smaller than the median of $x$ and $x_j$ larger than the median of $x$; $X_n$ has
been sorted such that $x_1 \leq x_2 \leq ... \leq x_n$. Following Hubert and Van der Veeken (2008), the outlier bounds are then modified
according to:

for  MC $\geq$ 0:

lower bound $= Q_1 - 2.0 \times e^{-4 \times MC} \times IQR$

upper bound $= Q_3 + 1.5 \times e^{3 \times MC} \times IQR$

for  MC < 0:

lower bound $= Q_1 - 1.5 \times e^{-3 \times MC} \times IQR$

upper bound $= Q_3 + 2.0 \times e^{4 \times MC} \times IQR$

In this study, the medcouple is calculated separately for each month, using all ozone values in that given month (for all years),
altitude, and latitude band using data that has previously passed rules 1 and 2. $MC$ is then used to define the outlier bounds,
and ozone values that lie outside these bounds are removed from the final product. Distributions of ozone for a selected number
of months, altitude, and latitude band are shown in Fig. 10 to Fig. 12; the median and outlier bounds of each distribution are
indicated as red and black dashed lines, respectively. Panel (a) of Fig. 10 to Fig. 12 shows the distribution of ozone values
before any screening was applied, while the panel (b) shows the distribution of the remaining data after the 200 % uncertainty



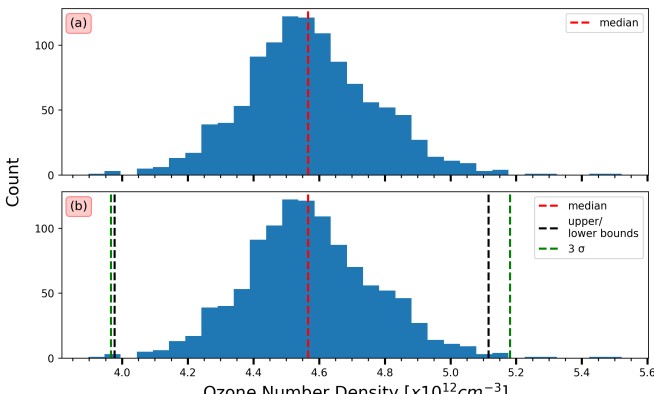

**Figure 10.** Probability density function of SAGE II ozone at 25 km between 25° and 35° S for July. (a) Probability density function of SAGE II ozone before any screening is applied to the data. (b) Probability density function of SAGE II ozone after the 200 % uncertainty rule and the aerosol LOS optical depth rule were applied. Red line indicates the median of the distribution shown in each panel, while the black dashed lines represent the upper and lower boundaries calculated as described in the text. As reference, panel (b) includes 3 times the standard deviation of the distribution ($3\sigma$) to represent the selection criteria similar to current rule #11. All data points that lie outside the black dashed lines will be removed from the data set. For more details see text.

rule and the aerosol LOS optical depth rule were applied. All data that lie beyond the boundaries (black lines in Fig. 10 to 12), will be removed from the data set. This new outlier rule (skewness rule) removes only large outliers and generally fewer data points from the distribution than the current rule #11 as outlier bounds (compare black dashed lines to green dashed lines in Fig. 10 to 12). It also generally passes the eye test of where empirically determined outlier bounds would be placed (unlike current rule #11) as shown in Fig. 10. Overall, less than 3 % of all data points are eliminated using this rule and the median of

the distribution is nearly unchanged (as denoted in Fig. 10 to 12), as all median values (red lines in each Fig.) are similar.

      The effect of the three screening rules for the SAGE II ozone data is shown on Fig. 13. Ozone at 17 km between 15° and 25° N is shown in Fig. 13a, while ozone at 20.5 km between 25° and 35° S is shown in Fig. 13b. At 17 km, majority of ozone (mainly negative or low ozone values) is removed by the 200 % uncertainty rule (about 6 %), followed by the outlier screening rule (about 3 %). The aerosol rule plays a significant role only during the Mt Pinatubo period. At higher altitudes, the 200 %

screening rule and the outlier rule become less important, and the majority of ozone is removed due to the aerosol rule (Fig. 13b), mainly during the Mt Pinatubo period. Above 30 km, very few ozone data are removed throughout the data record (not shown), and the higher you go the less important are the screening rules. Overall, the current rules eliminate up to 13 % of the ozone data in a altitude and latitude band primarily below 23 km, while the new screening recommendations developed in this study remove no more than 5 % of the data. While, at higher altitudes, the total number of data points that are eliminated by

the current and new rules is similar (less than 2 % depending on altitude and latitude band), they do not remove the same data points and differences are apparent which will be discussed in more detail in the next section.

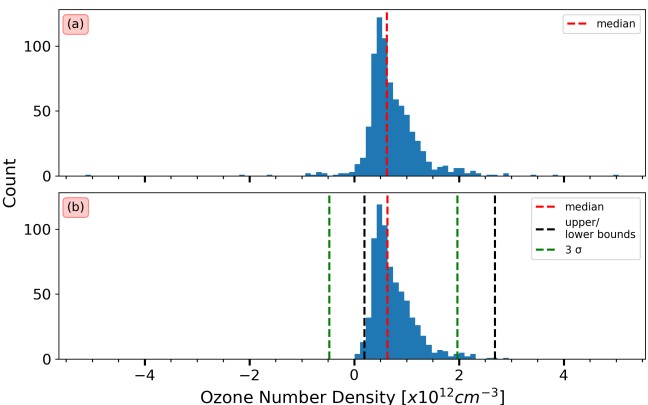

**Figure 11.** As Fig. 10 but for ozone data at 18 km between $15°$ and $25°$ N for January.

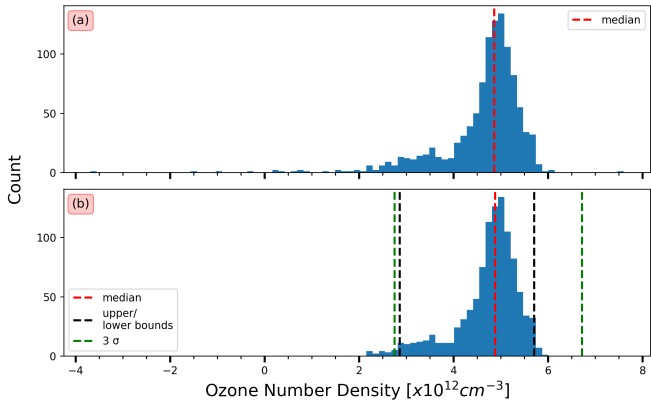

**Figure 12.** As Fig. 10 but for ozone data at 20 km between $65°$ and $75°$ S in December.

## 5.4 Comparing data rules

A comparison between the remaining SAGE II ozone data at a number of altitudes and latitude bands after the new and current rules have been applied is shown in Fig. 14. While, at 16 km, applying the new and current rules to the SAGE II ozone data between $5°$ S and $5°$ N latitude, remove about the same number of data points (46 % compared to 48 %), differences are apparent as shown in Table 1. About 14 % of the data that remain included in the screened ozone data set after the current rules were applied, are now removed when applying the new rules to the same data set. In this case, these additional data points that are removed are mostly the negative and low ozone values that were flagged as bad using the the 200 % relative uncertainty. On the other hand, about 16 % of the data that are removed when applying the current rules, are now retained when using the


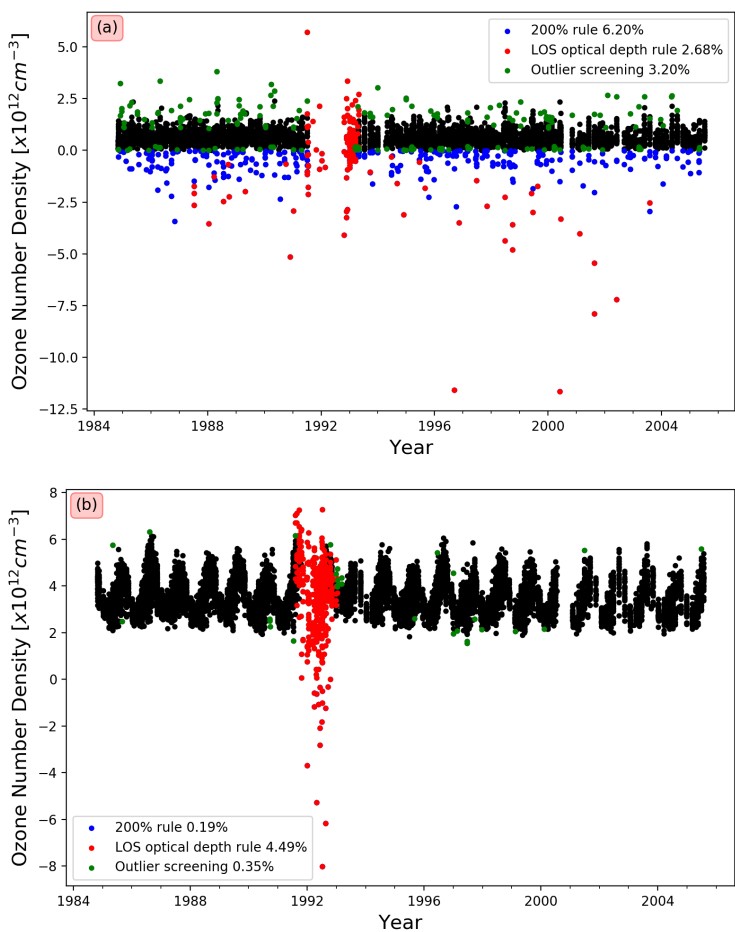

**Figure 13.** Number density ozone at 17 km between 15° and 25° N (a) and at 20.5 km between 25 and 35° S. The black dots represent the full SAGE II ozone data sets at the given altitude and latitude band, while the colored dots represent the ozone values that would be removed by applying the new rules (as indicated in the legend). Note that all rules were applied individually to all data points.





| 16 km between 5° S and 5° N | | | 20 km between 45° and 55° N | | 20.5 km between 65° and 75° N | |
|---|---|---|---|---|---|---|
| | Current incl. | Current removed | Current incl. | Current removed | Current incl. | Current removed |
| New incl. | 38.43 % | 15.74 % | 89.00 % | 7.84 % | 87.73 % | 11.08 % |
| New removed | 13.81 % | 32.02 % | 0.12 % | 3.04 % | 0.35 % | 0.84 % |

**Table 1.** Percentage of data removed/included depending on new and current rules.

new rules. The retained values include ozone data during the Nyamuragira/Nevado del Ruiz (1985/1986) eruptions and the Mt Pinatubo eruption; data which do not show obvious reasons on why they should have been removed. This suggests that the aerosol LOS optical depth rule is more appropriate than using a threshold for the aerosol extinction coefficient as it is used in the current set of screening rules. The differences between the data sets that remain after the new (Fig. 14a) and current rules (Fig. 14b) are substantial and could potentially lead to different mean values for the given altitude and latitude band.

At higher altitudes (20 km), the number of data points that remain after applying the new rules is higher than after applying the current rules. While about 89 % of the data in between 45° and 55° N and at 20 km altitude remain after the current rules were applied, more than 96 % of the data remain after the new rules were applied to the same data set. The current rules remove about 7 % more data, especially during the Mt Pinatubo period, than the new rules which retain a lot of ozone measurements during the Mt Pinatubo eruption period. Only about 0.1 % of the data that remained after the current rules were applied, are

removed when using the new data screening recommendations (Table 1). In this case, the new recommendations provide better means to eliminate biased ozone measurements, retaining important ozone measurements during the Mt Pinatubo eruption period.

The retention of data during the Mt Pinatubo period is even more pronounced at 20.5 km between 65° and 75° N (Fig. 14). The new rules only remove about 1 % of the overall data (mainly outliers, i.e. the skewness rule) in that latitude band, while the

current rules remove more than 11 % of the ozone measurements. This again indicates that the current rules are too restrictive when it comes to decide what a 'good' ozone measurement is, leading to unexpected results and removing valuable data during volcanic events such as the Nyamuragira/Nevado del Ruiz and Mt Pinatubo eruptions.

At higher altitudes, above 35 km, both set of rules retain more than 98.5 % of the ozone data, and above 40 km, more than 99 % of the data remain after any screening rule is applied. For example, at 35.5 km between 35° and 45° S, both set of rules

remove 98.99 %. The new outlier screening rule removes a total of 0.81 % of the data, while the current set of the rules remove 0.65 % of the data. In both cases, only the outlier screening rule removes any ozone data, and at higher latitudes, the new outlier screening rule removes a few more data points. As expected, these results indicate, that the measurements at the lower altitudes and in the mid-stratosphere are mostly effected by aerosol and clouds. Despite the low number of ozone data that are removed above 35 km, we recommend to apply the new set of screening recommendations at all levels, as outlier remain part

of the data set which need to be removed before any analysis of the data is performed.

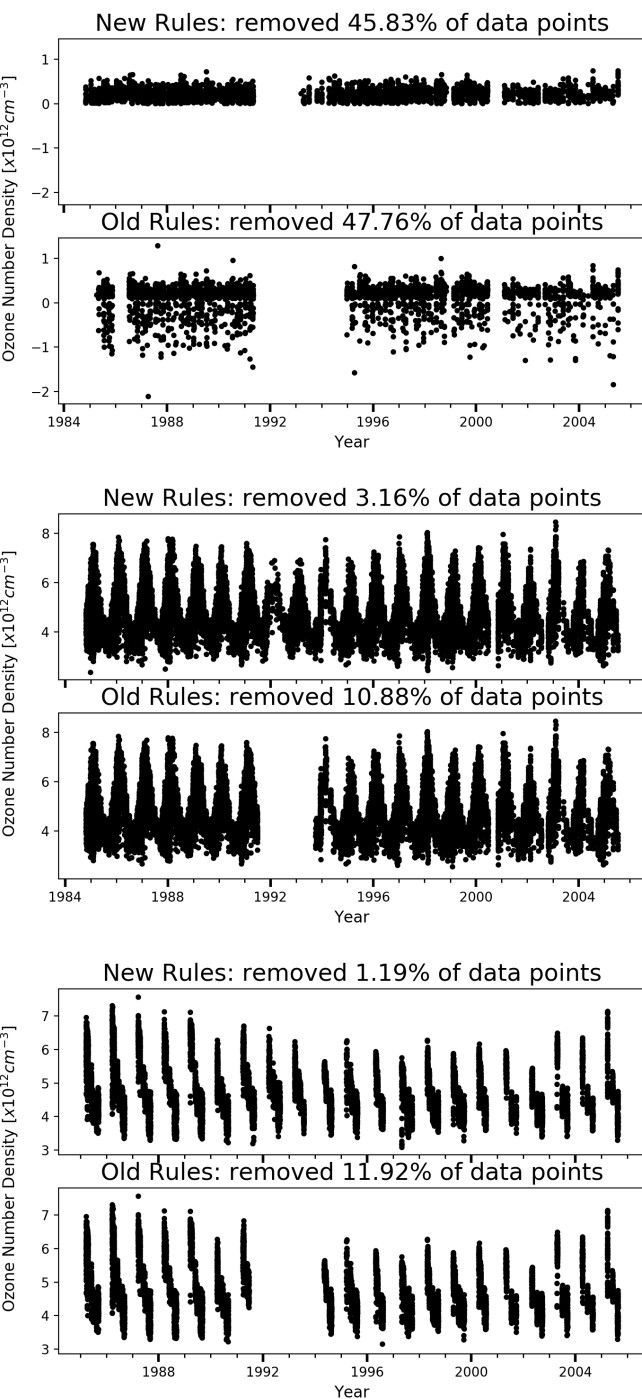

**Figure 14.** SAGE II ozone at: 16 km between 5° S and 5° N, 20 km between 45° and 55° N, and 20.5 km between 65° and 75° N. SAGE II data shown represent the remaining ozone data after the new (top) and current (bottom) rules were applied.

## 6 Conclusions

In this study, we developed adequate SAGE II ozone data screening rules, using only three rules compared to up to 11 rules used in previous similar efforts including screening used to produce homogenized ozone data sets (Davis et al., 2016; Hassler et al., 2018). The new rules are simple and everything required to apply these rules is provided with the SAGE II data available from the NASA data centre (ASDC). The new recommendations take into account how the measurements were made and are:

- Remove all ozone data when the corresponding aerosol LOS optical depth exceeds a value of 3.

- Remove all ozone data with an uncertainty of exactly 200 %

- Remove all data that fall outside boundaries calculated using the skewness test of a distribution.

In general, the new rules remove fewer data from the overall SAGE II ozone data set and the new rules are more robust and appropriate than previous versions particularly in handling non-aerosol related outliers (current rule #11) as it is not restricted to a normal distribution of the ozone values in a given latitude band.

SAGE II ozone data are used as the reference ('gold standard') to which other satellite data (such as AURA MLS, HALOE etc.) are adjusted to when generating a homogenised long-term ozone time series of monthly mean zonal means (Hassler et al., 2018; Davis et al., 2016). While the monthly mean zonal mean values calculated from the data set remaining after the current and new rules have been applied do not vary significantly, the impact on the homogenisation of several data set from different sources can be significant, as the data are adjusted individually. Having more SAGE II data available, and applying a more robust outlier screening, will most likely lead to a better adjustment of other satellite data to SAGE II.

*Data availability.* SAGE II ozone data set used in this paper, including calculated line-of-sight optical depths, are publicly available in NetCDF format from Zenodo at https://doi.org/10.5281/zenodo.3710518 and are distributed under the Creative Commons Attribution 4.0 International Public License. The complete SAGE II data set version 7.0 (binary format) is available free of charge from the NASA Atmospheric Science Data Center (ASDC): SAGE II Science Team (2012), SAGE II Version 7.00 Data, Hampton, VA, USA: NASA Atmospheric Science Data Center (ASDC), at (doi:10.5067/ERBS/SAGEII/SOLAR_BINARY_L2-V7.0)

*Author contributions.* SK wrote the paper with support and input from LT. SK and LT developed the new screening recommendation with porgamming support from LB.

*Competing interests.* The authors declare that they have no conflict of interest.





*Acknowledgements.* This work was funding in part through the Deep South National Science Challenge (CO1X1445). We would like to thank Robert Damadeo for some valuable feedback and helpful discussions. SAGE II version 7 data were obtained from the NASA Langley Research Center Atmospheric Science Data Center.





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
