# Peer review of "Simplified SAGE II ozone data usage rules"

_Earth System Science Data, 2020_

## Referee Comment (RC1) · Anonymous Referee #1 · 15 Apr 2020

Simplified SAGE II ozone data usage rules

S. Kremser et al.

This paper has been written with the primary goal of establishing a new set of screening criteria specifically for the version 7.00 ozone data distributed by the SAGE II team. Three new screening criteria are recommended by the authors and this list greatly simplifies the existing, somewhat complicated and seldom official set of recommendations. The new criteria developed by the authors, are more directly based on expert knowledge of the measurements and the measurement technique, and do not rely on visual inspection nearly as much as the previous set. Specifically, the new criteria are focused primarily on a more accurate, measurement-based discrimination of ozone measurement made in the presence of high aerosol loading. The new recommendations are evaluated against the old set and it is shown that less data are removed, and their re-

moval likely does not contribute to measurement biases in the same way as those that were created when the old criteria were applied.

General Comments

This is a very useful work and it will greatly simply the screening process for all scientists interested in using the historical SAGE II data set. I appreciate that even this long after SAGE II stopped producing valuable scientific data, the team associated with the instrument are still adding value through the improvement of this very important data record. I recommend this paper be published after addressing the comments below.

Specific and Mostly Minor Comments 1. There is inconsistent use of the nomenclature version 7.0 and 7.00 in the text. 2. (page 3, line 92) Can you better explain what horizontal resolution in square km means? 3. (page 6, line 172) The statement with "removing all points less than a fixed relative error ..." seems to be inconsistent with rule #3. Have I misunderstood? 4. (page 7, line 177) The two sentences that begin this paragraph seem to be repetitive. 5. (page 7, line 185) This probably should read "percentage may be greater ..." 6. (page 9, line 218) Can you clarify for me the statement about beta angles between -47 and 47 degrees? Aren't the low beta angles the good ones for making occultation measurements? 7. I think the discussion about "short events" should be moved to a position in the text immediately following the description of the current rules, as it is a "current rule". 8. (page 10) Are the cross sections for ozone and Rayleigh tabulated somewhere. At a minimum the ones used in the analysis presented should be referenced. 9. For Figure 4 is it possible to give an occultation identifier so a user can recreate the plot? In general if we the users want to employ these rules, we would benefit from some numerical examples in order to verify our calculations. 10. (page 13) The calculation of the aerosol extinction at 600 nm seems out of place. I suggest putting it very close within the text to the other relevant calculations. This is for ease of reference for somebody using the "recipe". 11. The 200% discussion seems a little odd. Was it the intention of the data producer (Damadeo I assume) to discard these measurements? If so, this should just be clearly

communicated. 12. (page 25) The figure caption, and the figure, needs to include a), b) and c) indicators to clearly distinguish the altitude and latitude bins. 13. (page 24, line 470) The statement beginning "In this case . . ." is perhaps demonstrated throughout the paper, but it is not proven. It is intuitive that the statement is correct but much more work needs to be done to prove it. I suggest softening the language around the statement. It's my opinion that you don't need to prove it for the paper. This goes for other similar statements sprinkled throughout the paper.

Summary

This paper should be published with only minor revisions. It is a valuable contribution that I will use almost immediately.

---

## Referee Comment (RC2) · Anonymous Referee #2 · 5 May 2020

I have read this nice manuscript. The authors have developed new screening recommendations for the SAGE II ozone data set and compared the new set of usage rules to the ones currently used. The new methods of data screening were designed based on how the measurements were made. With only three screening rules, the authors successfully removed the outlier values that are affected by high aerosol loadings and thus are not reliable. In my opinion, this paper is well written with data and methods described in a detailed and transparent way. The information given here is very helpful for the users of the SAGE II ozone data. I suggest the paper can be published as it is. Perhaps a suggestion: have more discussions on the limitations of the new data usage rules although they are already improved compared to the currently used ones, and try to give more suggestions on the future development of the SAGE II ozone data.

[Figure]

2020.

---

## Author Comment (AC1) · 15 May 2020

**Reply to Anonymous Referee #2**

We would like to thank both reviewers for their positive review. Point-by-point responses follow below, with the reviewer's comment repeated in blue and our response in black.

Suggestion: have more discussions on the limitations of the new data usage rules although they are already improved compared to the currently used ones, and try to give more suggestions on the future development of the SAGE II ozone data.

We added the following sentences to the end of the conclusion:
"The new SAGE II screening rules are still empirical rules based on our best scientific judgement and based on how the measurements were made. It is possible that applying these new rules are still throwing out data, particularly based on ozone anomalies, that are 'good' data and retaining ozone values that appear valid when they are 'bad'. It is incumbent on the user of this approach to apply them in a way that is consistent with their own requirements potentially easing or tightening the new data usage rules, perhaps in the more extreme periods as exemplified by the Mt. Pinatubo eruption."

As for a recommendation, we believe that based on the results presented in this paper, it would be valuable to re-examine the error calculations for ozone where 200% is recorded and used as a flag for anomalies. However, the SAGE II team is aware of this recommendation and therefore we decided not to include this as a recommendation in the paper.

---

## Author Comment (AC2) · 15 May 2020

**Reply to Anonymous Referee #1**

We would like to thank both reviewers for their positive review. Point-by-point responses follow below, with the reviewer's comment repeated in blue and our response in black.

*Specific and Mostly Minor Comments*

There is inconsistent use of the nomenclature version 7.0 and 7.00 in the text.

We have corrected the inconsistent use of the version numbering throughout the manuscript. We are now referring to version 7.00 to be consistent with the SAGE II release notes.

(page 3, line 92) Can you better explain what horizontal resolution in square km means?

For any given altitude, where SAGE II reports data, the total spatial spread (area) over which the measurements contribute is much larger than the simple line of sight through a layer (0.5 km by maybe 130 km). It is a number of these that move their location in the atmosphere due to the motion of the space craft. The size of the volume depends on the beta angle, i.e. the elevation angle of the Sun with respect to the orbital plane of the spacecraft. This is explained in detail in a publication by Thomason et al. (2003), which we now included as a reference. We cited the wrong paper at this point and now corrected that.

(page 6, line 172) The statement with "removing all points less than a fixed relative error..." seems to be inconsistent with rule #3. Have I misunderstood?

Here we wanted to describe that using the relative error as a screening criterion will lead to biases and that applies to all data. However, we agree that the statement as it is can lead to confusion as rule #3 only concerns ozone values below 35 km. We have clarified that in the paper by including the following:

"Removing ozone values that have an associated relative error greater than a fixed value (e.g. rule #3) will bias the remaining data set (Fig. 2), leading to a larger positive value."

(page 7, line 177) The two sentences that begin this paragraph seem to be repetitive.

We merged these two sentences into one to avoid repetition.

"To investigate the impact of each rule on the SAGE II ozone data, the usage rules outlined above were applied individually to the whole SAGE II ozone data set at a given altitude and within a given 10° latitude band."

(page 7, line 185) This probably should read "percentage may be greater..."

Corrected.

(page 9, line 218) Can you clarify for me the statement about beta angles between -47 and 47 degrees? Aren't the low beta angles the good ones for making occultation measurements?

After a series of battery issues, instrument events were shortened in how long they collected data in order to reduce the battery drain during an event. In error, the data collection time for

an event, particularly for low beta angle/short duration events, were shortened to an extent that damaged data quality (mostly a normalization issue). Once this issue was identified, event data collection times were lengthened sufficiently to eliminate this problem. As mentioned in the paper, this issue does not apply to version 7.00 since it has been corrected (see Damadeo et al. 2013). We just wanted to explain to the reader why we do not address the screening recommendation according to the beta angles which is widely used in other studies.

I think the discussion about "short events" should be moved to a position in the text immediately following the description of the current rules, as it is a "current rule".

That is a good suggestion and we moved the discussion of the 'short-even' rule to the end of section 2.

(page 10) Are the cross sections for ozone and Rayleigh tabulated somewhere. At a minimum the ones used in the analysis presented should be referenced.

We now included the values that we used in our calculations.

For Figure 4 is it possible to give an occultation identifier so a user can recreate the plot? In general if we the users want to employ these rules, we would benefit from some numerical examples in order to verify our calculations.

We clarified the data source of the profiles shown in Figure 4 in the caption in the revised manuscript.

(page 13) The calculation of the aerosol extinction at 600 nm seems out of place. I suggest putting it very close within the text to the other relevant calculations. This is for ease of reference for somebody using the "recipe".

The calculation of the aerosol extinction at 600 nm required the Ångström coefficient, α that is described here. That is why we prefer to leave the calculation of k_a_600 here but we added a sentence to clarify why the equation is shown at this point in the paper.

"α together with the aerosol extinction coefficient at 1020 nm is then used to determine the aerosol extinction coefficient at 600 \unit{nm}, following: …"

The 200% discussion seems a little odd. Was it the intention of the data producer (Damadeo I assume) to discard these measurements? If so, this should just be clearly communicated.

The origins and intention of introducing the 200% relative uncertainty by the SAGE II team was to flag negative ozone values that had low but unrealistic uncertainties (as described in the paper) in the lower atmosphere. The filtering of the SAGE II data according to the 200% uncertainty has not typically been part of the standard usage rules but was described and recommended in the SAGE II v7.00 release notes. In the release notes, they simply recommended to exclude all lower altitude data points where the uncertainty was >=200%, which Rob Damadeo (i.e. a data producer) has informed us was a "simple yet overly conservative way" to flag these anomalies. However, we believe from our analysis that only these artificially flagged data points, which have **exactly** 200% uncertainty, should be removed.

 The figure caption, and the figure, needs to include a),b) and c) indicators to clearly distinguish the altitude and latitude bins.

We implemented that change.

 The statement beginning "In this case..." is perhaps demonstrated throughout the paper, but it is not proven. It is intuitive that the statement is correct but much more work needs to be done to prove it. I suggest softening the language around the statement. It's my opinion that you don't need to prove it for the paper. This goes for other similar statements sprinkled throughout the paper.

We softened the language a little.